# A comprehensive outline of antimicrobial resistance, antibiotic prescribing, and antimicrobial stewardship in South Africa: A scoping review protocol

**Suwayda Ahmed** [ID]*, **Rukshana Ahmed, Razia Zulfikar Adam**

Department of Prosthodontics, Faculty of Dentistry, University of the Western Cape, Cape Town, South Africa

* suahmed@uwc.ac.za

**Data Availability Statement:** Data generated or analysed in this study is included in this published article (and its supplementary information).

## Abstract

### Introduction

The global prevalence of antimicrobial resistance transcends geographical and economic boundaries, affecting populations worldwide. Excessive and incorrect use of antibiotics encourages antimicrobial resistance which leads to complex treatment strategies for infectious diseases and possible failure of treatment. The incorrect and unnecessary prescribing of antibiotics places a burden on healthcare costs and thus, antimicrobial resistance is evident globally as a major public health concern. The impact is particularly pronounced in low to middle-income countries, where limited access to healthcare exacerbates the crisis. This scoping review aims to comprehensively map the evidence of antimicrobial resistance in healthcare settings, encompassing the exploration of antibiotic prescribing practices and the implementation of antimicrobial stewardship initiatives in South Africa.

### Methodology

This protocol has been registered in the Open Science Framework (https://doi.org/10.17605/OSF.IO/PWMFB). This review will consider all types of study designs, conducted within South Africa. Studies that are published in English for the period 2019–2024, and that explore antimicrobial resistance (AMR) evidence in healthcare in South Africa, including antibiotic prescribing trends and antimicrobial stewardship and surveillance initiatives will be included. Non-English publications, studies outside of South Africa, animal and environmental studies will be excluded. The criteria for the scoping review will be set by two reviewers. The Preferred Reporting Items for Systematic Reviews and Meta-Analyses for Scoping Reviews (PRISMA-ScR) tool will be used. Studies will be identified through an extensive search of peer-reviewed and grey literature databases. The results of the review will be tabulated and include a narrative synthesis of the findings.

**Funding:** The author(s) received no specific funding for this work.

**Competing interests:** The authors have declared that no competing interests exist.

**Abbreviations:** AMR, antimicrobial resistance; LMIC, Low to middle income countries; PCC, Population-Concept-Context; PRISMA-ScR, Preferred Reporting Items for Systematic Reviews and Meta-Analysis; WHO, World Health Organization.

## Introduction

The World Health Organization (WHO) has highlighted antimicrobial resistance (AMR) as a life-threatening global health threat [1]. AMR is driven by various factors, of which the overuse of antibiotics is the most dominant [2]. Inappropriate prescribing and excessive use of antibiotics are a major driver of AMR worldwide [3–6]. In 2019, antimicrobial drug resistance infections contributed to 4.95 million deaths globally, with 1.27 million deaths attributable to bacterial AMR. Low to middle income countries (LMICs), especially in Sub-Saharan Africa are especially vulnerable to the threat of AMR [2]. Between 2000 and 2015, LMICs have seen a substantial increase in AMR rates due to the lack of access to clean water, sanitation, and hygiene [1]. Drivers of AMR in LMICs are influenced by political, economic, socio-cultural, and ecological factors that influence the profile of these countries [7].

Managing AMR in low- and middle-income countries, such as those in Africa prove challenging when compared to their high-income counterparts. Sub-Saharan African countries efforts to implement effective and workable AMR stewardship programs are often disordered by various factors; which include lack of human resources, decreased investment, decreased infrastructural and institutional capacities [8, 9]. Lack of data around AMR complicates the management of AMR in Sub Saharan Africa [9]. Country specific data is not routinely assembled and is often not shared with national regulatory bodies. This places limitations on the ability of these regulatory bodies to effect national action [10, 11].

South Africa, which is categorised as a middle-income country is not exempt of the threat of AMR, which manifests itself in increased mortality rates, extended hospitalization, escalated cost of healthcare and expenditure of healthcare resources [12, 13].

Addressing the threat of AMR requires targeted engagement. This includes action which is aimed at the prevention of infection and the management of microbial transmission, improvement of antibiotic use via surveillance, as well as use of prescribing guidelines amongst prescribers and antimicrobial stewardship methods to stop the spread of resistant organisms [14].

In recognizing the threat of AMR, especially as it relates to LMICs, the National Department of Health in South Africa drafted the National AMR Strategy Framework [15]. This framework aligns with the WHO Global Action Plan as it underscores interdisciplinary efforts, antimicrobial stewardship and the elevation of infection prevention and control [16]. The aim of this national strategy framework is to provide guidance in order to limit the increase of antibacterial resistance and improve the standard of health amongst the general population. One of the strategic objectives of the strategy framework is to promote the accurate use of antibiotics in humans, and thus introduce antimicrobial stewardship at institutional levels. This would include antimicrobial stewardship (AMS) programmes at every health care facility, and district to regulate appropriate prescribing. In addition, reduction of antibiotic prescribing and workforce development also features amongst the objectives of the national strategy framework. The emphasis here is to facilitate the education of AMR through teaching and continuous training exercises [15].

The South African healthcare system is varied, and consists of a public health sector which is funded by the national government and a private health sector which is funded via medical insurance or private individuals. The majority of the South African population rely on the public healthcare sector to access healthcare [17–20]. The lack of resources, limited microbiological laboratory testing, delay in feedback of results and poor infrastructure are factors which hinder the accurate description of AMR rates in South Africa [9, 21]. Healthcare facilities in South Africa are at varying phases of antimicrobial surveillance and stewardship. Studies confirmed that although healthcare facilities and prescribers adhered to antibiotic prescribing guidelines, inappropriate prescribing is still evident amongst South African prescribers [21–23].

### Rationale for scoping review

South Africa, which is categorised as a middle-income country in Sub-Saharan Africa, faces significant challenges regarding the public health crisis of AMR. Antimicrobial stewardship practices vary across healthcare sectors despite the presence of guidelines, contributing to inappropriate prescribing behaviours. The National Department of Health has identified prevalent pathogens responsible for resistance patterns among South African populations, guiding local efforts in antimicrobial surveillance and stewardship [23]. Initial literature searches have confirmed existing reviews on antimicrobial stewardship and AMR across Africa [10, 24, 25]. This scoping review is aimed at taking an all-encompassing view at AMR rates and evidence, antibiotic prescribing trends among prescribers as well as surveillance measures and stewardship programmes in place in South Africa.

## Methods and analysis

The key research question leading this review is: What is known from the literature about AMR in healthcare in South Africa?

### Review questions

What is the current status of AMR in South Africa; incorporating AMR awareness, antibiotic prescribing practices and antimicrobial stewardship?

### Methodology

The reviewers (SA and RA) will follow the Preferred Reporting Items for Systematic Reviews and Meta-Analysis (PRISMA-ScR) approach for selection of articles on the current AMR situation, antibiotic knowledge and prescribing patterns and management of AMR [26]. The methodological framework by Arksey and O'Malley will be used to conduct this scoping review [27]. The following steps will be followed: identifying a clear research question; identifying related studies in literature; selection of articles; data extraction; summarise, synthesise and report on findings.

### Search strategy

A comprehensive search strategy will be developed, using the following databases for peer-reviewed full text articles: PubMed, Scopus, Wiley, Directory of Open Access Journals and Science Direct and the WHO database. The Google search engine will be used to search for grey literature.

The following terms will be used in a combination of key Medical Subjects Heading (MeSH) terms and Boolean operators ("OR/AND"):

(Antibiotics OR antimicrobials) AND (Antimicrobial OR antibacterial resistance) AND Antimicrobial Stewardship AND Antibiotic Prescribing AND South Africa AND Healthcare

**Inclusion criteria.**  Full-text peer-reviewed journal articles including study designs (quantitative, qualitative, observational, mixed methods studies) which are published in English, grey literature, and human studies for the period of 2019–2024.

**Exclusion criteria.**  Non-English articles, systematic reviews, narrative reviews, studies outside of South Africa, animal and environmental studies.

A comprehensive search strategy will be developed for databases and grey literature. S1 Appendix demonstrates the search strategy created in PubMed.

## Study selection

The two reviewers (SA and RA) will independently search through the literature, and assess the suitability of the studies from the databases as determined by the eligibility criteria. Study eligibility and data extraction forms will be used to guide the researchers to select the suitable documents and extract data independently, which will be screened manually. The two sets of literature will be compared. Any duplicate studies will be excluded manually. Studies where the study abstract does not relate to the focus question will be excluded. The two reviewers will independently manually screen titles and abstracts based on the predetermined inclusion and exclusion criteria. Thereafter, the two reviewers will independently assess the full text articles to determine the final eligibility. Any differing opinions will be resolved by consultation with the second reviewer, if required. A third reviewer (RZA) will assist if consensus cannot be reached. Data extraction will be completed from the final full-text included documents obtained from the different databases (Science Direct, Scopus, PubMed, Wiley, Directory of Open Access Journals; and health organizations such as the World Health Organization (WHO).

## Identification of relevant studies

The inclusion criteria for this scoping review will be based on the Population-Concept-Context (PCC) framework (Table 1).

The population of interest in this scoping review will be healthcare workers and undergraduate healthcare students at healthcare facilities in South Africa. The concept will consist of evidence of AMR, awareness and knowledge of AMR among healthcare workers and students, antibiotic prescribing practices among healthcare workers, antimicrobial surveillance measures and antimicrobial stewardship interventions in health facilities.

## Data extraction and charting

The data will be organized and reported on, according to themes as described in Table 2.

Data from quantitative, qualitative, cross-sectional studies, observational studies will be managed systematically. Data from quantitative studies will be summarised in a descriptive format, in an attempt to understand trends, patterns, and findings from the data that are relevant to AMR in South Africa. A thematic analysis will be used for qualitative data in observational, intervention, and cross-sectional studies where data will be identified, categorized, and summarized narratively. All data will be extracted using a standardised data extraction form (S2 Appendix), and the data will be charted according to the predetermined themes. The data will then be charted according to the themes of AMR evidence, antibiotic prescribing, stewardship and surveillance, and healthcare worker knowledge. The final results will be presented in a narrative and descriptive format; with an emphasis on the key trends and gaps in the current evidence on AMR in healthcare in South Africa.

**Table 1. Arksey and O'Malley (2005) framework for eligibility of the research question.**

| Criteria | Determinants |
|---|---|
| Population | Healthcare workers, undergraduate students and healthcare facilities |
| Concept | Evidence, knowledge and awareness of AMR |
|  | Antibiotic prescribing practices |
|  | Stewardship and surveillance of AMR |
| Context | Healthcare in South Africa |

**Table 2. Themes for data extraction and charting.**

| Key Themes | Sub-themes |
|---|---|
| The current antimicrobial resistance situation | • AMR evidence<br>• Knowledge and awareness of AMR among HCWs and students<br>• Perception regarding rational prescribing of antimicrobials in relation to AMR |
| Antibiotic knowledge, prescribing patterns and guidelines | • Antibiotic prescribing patterns<br>• Knowledge of antimicrobial agents<br>• Compliance with antibiotic prescribing guidelines |
| Management of AMR (stewardship and surveillance) | • Stewardship policies<br>• Surveillance mechanisms<br>• Knowledge and awareness of management of AMR (stewardship and surveillance) among HCWs and students |

**Screening of literature.** Two reviewers (SA and RA) will independently screen titles and abstracts against the predetermined inclusion and exclusion criteria.

**Data extraction.** Relevant data will be extracted manually by reviewers (SA and RA), using a standardized data extraction form. The form includes information such as study title, year, study design, location, themes addressed and key findings. The reviewers will extract data independently to ensure accuracy, transparency and consistency.

**Data charting.** Data extracted will be charted manually, and arranged according to themes and sub-themes as identified in Table 2.

## Summarising and reporting results

The results will be outlined in a logical and descriptive summary that align with the objective/s of this review. The results will be mapped to the thematic framework as well as in descriptive form. The findings will be narratively summarised.

## Ethics approval

Ethical approval for this project was obtained from the Biomedical Research Committee at the University of the Western Cape, (BMREC Reference Number: BM24/4/6).

## Supporting information

**S1 Appendix. Search strategy PubMed.**
(DOCX)

**S2 Appendix. Data extraction table template.**
(XLSX)

**S1 Checklist. PRISMA-P-SystRev-checklist.**
(DOCX)

## Author Contributions

**Conceptualization:** Suwayda Ahmed.

**Methodology:** Suwayda Ahmed, Rukshana Ahmed.

**Supervision:** Razia Zulfikar Adam.

**Writing – original draft:** Suwayda Ahmed.

**Writing – review & editing:** Suwayda Ahmed, Rukshana Ahmed, Razia Zulfikar Adam.

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
