## [Decision Letter · Decision Letter 0]

7 Oct 2024

PONE-D-24-33778A Comprehensive Outline of Antimicrobial Resistance, Antibiotic Prescribing, and Antimicrobial Stewardship in South Africa: A Scoping Review ProtocolPLOS ONE

Dear Dr. Ahmed,

Thank you for submitting your manuscript to PLOS ONE. After careful consideration, we feel that it has merit but does not fully meet PLOS ONE’s publication criteria as it currently stands. Therefore, we invite you to submit a revised version of the manuscript that addresses the points raised during the review process.

We look forward to receiving your revised manuscript.

Kind regards,

Mabel Kamweli Aworh, DVM, MPH, PhD. FCVSN

Academic Editor

PLOS ONE

Reviewers' comments:

Reviewer's Responses to Questions

**Comments to the Author**

1. Is the manuscript technically sound, and do the data support the conclusions?

Reviewer #1: Partly

Reviewer #2: Yes

Reviewer #3: Partly

2. Has the statistical analysis been performed appropriately and rigorously? 

Reviewer #1: No

Reviewer #2: Yes

Reviewer #3: N/A

3. Have the authors made all data underlying the findings in their manuscript fully available?

Reviewer #1: No

Reviewer #2: Yes

Reviewer #3: No

4. Is the manuscript presented in an intelligible fashion and written in standard English?

Reviewer #1: Yes

Reviewer #2: Yes

Reviewer #3: Yes

5. Review Comments to the Author

Reviewer #1: I believe this should be a protocol instead of research article, as I understand that the main article is still currently on and not submitted for publication yet, because most of the work results has not been presented, so you might want to change it from the online submission form or kindly clarify.

Reviewer #2: I appreciate the effort of the authors, just a few concerns about methodology to be addressed:

Methods

Search strategy: Provide the comprehensive search strategy for all databases used.

Study selection: Will any software (such as Rayyan or Covidence) be used for article screening?

Will any reference manager software used for deduplication of articles?A software should be used according to updated JBI Synthesis methodology. Please include these details.

See the following references for guidance

Peters, M. D. J., Marnie, C., Tricco, A. C., Pollock, D., Munn, Z., Alexander, L., McInerney, P., Godfrey, C. M., & Khalil, H. (2020). Updated methodological guidance for the conduct of scoping reviews. JBI Evidence Synthesis, 18(10), 2119–2126. https://doi.org/10.11124/JBIES-20-00167

- Peters, M. D. J., Godfrey, C., McInerney, P., Khalil, H., Larsen, P., Marnie, C., Pollock, D., Tricco, A. C., & Munn, Z. (2022). Best practice guidance and reporting items for the development of scoping review protocols. JBI Evidence Synthesis, 20(4), 953–968. https://doi.org/10.11124/JBIES-21-00242

Reviewer #3: Thank you for the opportunity to review your manuscript. I have provided some recommendations that I hope the authors find helpful.

Introduction

Lines 102-105: A recommendation is to put this information in a paragraph format, as a continuation of line 101.

Methods and analysis

Review question/research question: What is the difference between the review question and the research question? Lines 141-143 outlines the review question, while Lines 151-152 outline the key research question. Please clarify the difference between these two types of questions for the benefit of the reader. Research questions typically tend to be at the end of the introduction or background section. A recommendation is to include the research questions for the scoping review at the end of the introduction section.

Search Strategy: There is a need for more information in the inclusion criteria. Studies published within what timeframe are considered for inclusion in the study? The abstract mentions the consideration of all types of study designs for inclusion in the review. However, the body of the scoping review makes no mention of the study designs considered for inclusion. What specific types of study designs would be included in the review? How would the reviewers manage including data from different types of studies? For example, how would data from quantitative and qualitative studies; as well as data from cross-sectional studies and systematic reviews/meta-analysis be managed and analyzed for reporting?

Boolean terms: Would it be helpful to have a subsection for the Boolean terms used for the search strategy as a follow up to the search strategy section?

Literature screening, data extraction and charting: What platform would be used for screening selected papers? Information about key elements in the data extraction form would look better in a paragraph format, rather than in the numbered format. See Lines 195-199 for reference.

Abbreviations: Cross-check if PRISMA-SCR should be written as PRISMA-SCR or PRISMA-ScR. The formatting of the abbreviations and their respective full meanings do not follow the same format. Format LMIC, PCC, PRISMA-SCR and WHO to follow the same format as AMR. See Lines 205-207 for reference.

Appendix: Cross-check the respective PRISMA-SCR checklists to ensure that the corresponding items have the same correct page numbers.

Thank you for the opportunity to review this manuscript, I hope the authors find the feedback helpful.

6. PLOS authors have the option to publish the peer review history of their article (what does this mean?). If published, this will include your full peer review and any attached files.

Reviewer #1: **Yes: **Abdulhakeem Binhambali

Reviewer #2: **Yes: **Olubunmi Margaret Ogbodu

Reviewer #3: No

---

## [Author Response · Author response to Decision Letter 0]

4 Nov 2024

Good day 

Thank you for the valuable feedback. I have edited manuscript to accommodate all requirements and feedback, Please see my covering letter for detailed explanation of the edits and changes.

Thank you once again.

Kind regards

---

## [Decision Letter · Decision Letter 1]

19 Nov 2024

PONE-D-24-33778R1A Comprehensive Outline of Antimicrobial Resistance, Antibiotic Prescribing, and Antimicrobial Stewardship in South Africa: a scoping review protocolPLOS ONE

Dear Dr. Ahmed,

Thank you for submitting your manuscript to PLOS ONE. After careful consideration, we feel that it has merit but does not fully meet PLOS ONE’s publication criteria as it currently stands. Therefore, we invite you to submit a revised version of the manuscript that addresses the points raised during the review process.

We look forward to receiving your revised manuscript.

Kind regards,

Mabel Kamweli Aworh, DVM, MPH, PhD. FCVSN

Academic Editor

PLOS ONE

Journal Requirements:

Reviewers' comments:

Reviewer's Responses to Questions

**Comments to the Author**

1. Does the manuscript provide a valid rationale for the proposed study, with clearly identified and justified research questions?

Reviewer #1: Yes

Reviewer #3: Yes

2. Is the protocol technically sound and planned in a manner that will lead to a meaningful outcome and allow testing the stated hypotheses?

Reviewer #1: Yes

Reviewer #3: Partly

3. Is the methodology feasible and described in sufficient detail to allow the work to be replicable?

Reviewer #1: Yes

Reviewer #3: No

4. Have the authors described where all data underlying the findings will be made available when the study is complete?

Reviewer #1: Yes

Reviewer #3: No

5. Is the manuscript presented in an intelligible fashion and written in standard English?

Reviewer #1: Yes

Reviewer #3: Yes

6. Review Comments to the Author

You may also provide optional suggestions and comments to authors that they might find helpful in planning their study.

Reviewer #1: This protocol outlines a scoping review aiming to map the evidence on antimicrobial resistance (AMR), antibiotic prescribing practices, and antimicrobial stewardship (AMS) initiatives within South Africa. It aims to address the current state of AMR awareness, antibiotic prescribing patterns, and the implementation of AMS programs in healthcare settings. While the protocol offers a comprehensive approach, several areas could also benefit from further clarification, improvement, and refinement to strengthen the review's overall quality.

Although, the review addresses a highly pertinent global health issue, especially in low- and middle-income countries (LMICs), like South Africa and the aims and objectives of the review are clearly articulated, with a focus on key aspects of AMR in South Africa, including prescribing practices and stewardship efforts however, the inclusion and exclusion criteria (e.g., “studies published in English”) could be better justified. While it is understandable that language barriers may limit accessibility, non-English studies could offer valuable insights, especially in multilingual settings like a Nation as South Africa. It would be beneficial to include a rationale for this exclusion.

Also, excluding animal and environmental studies might be overly restrictive. Many environmental and veterinary studies contribute to understanding AMR dynamics in healthcare settings, especially in LMICs. The authors should provide a more robust argument for excluding these studies or consider including them for a more holistic perspective.

Reviewer #3: Thank you again for the opportunity to review your manuscript. Some feedback and questions from the previous review were not addressed. I reiterated some content from the previous review that were not addressed and provided additional feedback on the updated version of the manuscript.

Abstract

Line 75: It is important to define abbreviations before they are used. Please define AMR before using it.

Line 79: Define PRISMA before the use of the abbreviation.

Introduction

Line 117: Define AMS before using the abbreviation.

Methods and analysis

Lines 131-147: Previous feedback about review question/research question have not been addressed. There are research/review questions in two sections of the manuscript. What is the difference between the review question and the research question? Please clarify the difference between these two types of questions for the benefit of the reader. What are the author’s thoughts about including all the questions in one section? Research questions typically tend to be at the end of the introduction or background section. A recommendation is to include the review/research questions for the scoping review at the end of the introduction section. The rationale for scoping review section could stay in position as the final part of the introduction section, following which all the questions from lines 131-132 and 146-147 could be presented.

Line 156-164: Search strategy. Previous feedback about the search strategy has not been addressed. There is a need for more information in the inclusion and exclusion criteria. Would the year of publication be part of the search criteria? What specific types of study designs would be included in the scoping review? How would the reviewers manage including data from different types of studies? For example, how would data from quantitative and qualitative studies; as well as data from cross-sectional studies and systematic reviews/meta-analysis be managed and analyzed for reporting?

Line 163-164: Would “healthcare workers” referenced in Line 191, rather than “healthcare” currently included as a Boolean term be used as a Boolean term for the literature search?

A recommendation for smoother flow in reading information in the paper is to include the inclusion and exclusion criteria as well as the identification of relevant studies in the search strategy section. The study selection section can come after this section and lead to the data extraction and charting section.

Previous feedback about the literature screening, data extraction and charting have not been addressed. What tool, application or platform would be used for screening data extraction of the selected papers?

Data extraction and charting

Lines 200-204: Information about key elements in the data extraction form would look better in a paragraph format, rather than in the numbered format.

Thank you again for the opportunity to review your manuscript. I hope the authors find this feedback helpful.

7. PLOS authors have the option to publish the peer review history of their article (what does this mean?). If published, this will include your full peer review and any attached files.

Reviewer #1: **Yes: **ABDULHAKEEM BINHAMBALI

Reviewer #3: No

---

## [Author Response · Author response to Decision Letter 1]

13 Dec 2024

Thank you for your feedback and recommendations regarding our submitted manuscript. The additional insights have been helpful in managing the requested revisions. All feedback and the revisions implemented in response to the peer reviewers' comments have been tabulated in our rebuttal letter.

Kindest regards

---

## [Editor Report · Decision Letter 2]

17 Dec 2024

A Comprehensive Outline of Antimicrobial Resistance, Antibiotic Prescribing, and Antimicrobial Stewardship in South Africa: a scoping review protocol

PONE-D-24-33778R2

Dear Dr. Ahmed,

We’re pleased to inform you that your manuscript has been judged scientifically suitable for publication and will be formally accepted for publication once it meets all outstanding technical requirements.

Kind regards,

Mabel Kamweli Aworh, DVM, MPH, PhD. FCVSN

Academic Editor

PLOS ONE

Additional Editor Comments (optional):

Page 10: lines 214-215 - Please replace this sentence "Two reviewers (SA and RA) will be screen titles and abstracts independently, against the predetermined inclusion and exclusion criteria" WITH "Two reviewers (SA and RA) will independently screen titles and abstracts against the predetermined inclusion and exclusion criteria".
---

## [Editor Report · Acceptance letter]

17 Jan 2025

PONE-D-24-33778R2 

PLOS ONE

Dear Dr. Ahmed, 

I'm pleased to inform you that your manuscript has been deemed suitable for publication in PLOS ONE. Congratulations! Your manuscript is now being handed over to our production team.

Kind regards, 

on behalf of

Dr. Mabel Kamweli Aworh 

Academic Editor

PLOS ONE